# De-coupled NeuroGF for Shortest Path Distance Approximations on Large Terrain Graphs

**Samantha Chen** [1]  **Pankaj K. Agarwal** [2]  **Yusu Wang** [3]

## Abstract

The ability to acquire high-resolution, large-scale geospatial data at an unprecedented using LiDAR and other related technologies has intensified the need for scalable algorithms for terrain analysis, including *shortest-path-distance* (SPD) queries on large-scale terrain digital elevation models (DEMs). In this paper, we present a *neural data structure* for efficiently answering SPD queries approximately on a large terrain DEM, which is based on the recently proposed neural geodesic field (NeuroGF) framework (Zhang et al., 2023)— the state-of-the-art neural data structure for estimating geodesic distance. In particular, we propose a decoupled-NeuroGF data structure combined with an efficient two-stage mixed-training strategy, which significantly reduces computational bottlenecks and enables efficient training on terrain DEMs at a scale not feasible before. We demonstrate the efficacy of our approach by performing detailed experiments on both synthetic and real data sets. For instance, we can train a small model with around 70000 parameters on a terrain DEM with 16 million nodes in a matter of hours that can answer SPD queries with 1% relative error in at most 10ms per query.

## 1. Introduction

With rapid advances in LiDAR and related technologies, high-resolution, large-scale digital elevation models of terrains are being generated and made publicly available at an unprecedented rate. The tremendous opportunities provided by these data sets will, however, not be realized without availability of simple, scalable algorithms for various terrain-analysis tasks. This has led to extensive work on developing efficient algorithms for many terrain-analysis tasks such as visibility computation, hydrology analysis, mobility analysis, and answering proximity queries. In this paper, we focus on answering *shortest-path distance* (SPD) queries (also referred to as *geodesic distance* queries) on terrains. Throughout this paper we assume that a terrain is represented as a $xy$-monotone triangulated surface $\Sigma$ in $\mathbb{R}^3$. Many applications call for answering many SPD queries on the same terrain, so it is desirable to pre-process the input terrain into a data structure so that SPD queries can be answered quickly. The key metrics associated with such an approach are the trade-off between the size and the query time of the data structure as well as the preprocessing time.

Despite a rich body of literature on this topic, the SPD query problem on terrains is far from being solved both from theoretical and practical perspectives. For instance, a state-of-the-art algorithmic approach in a recent paper by (Wei et al., 2022) takes more than 200 hours to build a data structure of size around 1000MB for a terrain with 180K nodes to support approximate SPD queries, between pairs of vertices on the terrain, with relative error at most 0.1 in $10ms$ per query. Furthermore, it is hard to scale the current approaches to terrain DEMs with millions of nodes.

In view of the remarkable success of deep learning methods across various domains such as vision, graphics, robotics and GIS, it is natural to ask whether one can develop an effective *neural data structure* to answer SPD-queries efficiently on large-scale terrain DEMs. Precisely, given a terrain DEM $\Sigma$, we ask whether we can train a neural network that can accurately and efficiently estimate the geodesic distance between any two points on $\Sigma$. In this paper, we explore the design space of neural data structures for answering SPD-queries and propose a method that is scalable to large terrain DEMs.

**Related work.** Motivated by applications in robotics and GIS, algorithms for computing shortest paths on polyhedral surfaces in $\mathbb{R}^3$ have been studied extensively over the last four decades. The best-known algorithm for computing a shortest path between two given points on a terrain $\Sigma$ takes $O(n^2)$ time (Xin & Wang, 2009; Chen & Han, 1990;

---

*Equal contribution  [1]University of California - San Diego, Department of Computer Science and Engineering  [2]Duke University, Department of Computer Science  [3]University of California - San Diego, Halıcıoğlu Data Science Institute. Correspondence to: Samantha Chen <sac003@ucsd.edu>.

*Proceedings of the $42^{nd}$ International Conference on Machine Learning*, Vancouver, Canada. PMLR 267, 2025. Copyright 2025 by the author(s).

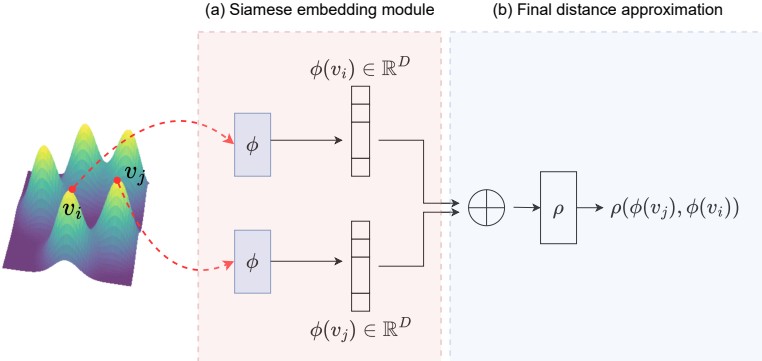

*Figure 1.* High level pipeline of NeuroGF (Zhang et al., 2023). Part (a): Siamese embedding module. Part (b): the distance approximation module. We propose to decouple the training of these two stages by first training the Siamese embedding module, freezing the trained Siamese network, and optimizing only the final distance approximation module.

Mitchell et al., 1987; 2000), where $n$ is the number of vertices on $\Sigma$. Although the running time can be improved to $O(n \log n)$ time (Schreiber & Sharir, 2008) for convex surfaces, no subquadratic algorithm is known for nonconvex surfaces. This has led to faster approximation algorithms for this problem: a subquadratic $(1 + \varepsilon)$-approximation algorithm was proposed in (Varadarajan & Agarwal, 2000). Subsequently, more approximation algorithms were developed in(Aleksandrov et al., 1998; Lanthier et al., 2001), whose run time is near linear under some assumptions on the geometry of the terrain, e.g., if the minimum angle of any face is bounded from below by a constant. There also has been much work in computational geometry, database, and GIS communities on developing practical algorithms for computing shortest paths on terrains; see (Kaul et al., 2013; 2015; Tran et al., 2020; Wei et al., 2022; 2024; Hazel et al., 2008) and references therein. Most of these approaches choose a large set of points on the terrain and construct a graph $G$ on the points so that the shortest-path distance on $G$ approximates the geodesic distance on the original terrain.

In machine learning, SPD queries are related to deep metric learning, where given a collection of pairwise distance between points sampled from a certain space $X$, one aims to train a neural network that can accurately estimate the metric $d_X : X \times X \to \mathbb{R}$ on $X$ (Ghojogh et al., 2022). Metric learning is often done via a *Siamese network* where given $x, y \in X$, each point is transformed by an identical neural network $\mathcal{N}$ to produce high-dimensional embeddings and the distance $\|\mathcal{N}(x) - \mathcal{N}(y)\|_p$ is then returned as an estimation for $d_X(x, y)$.

Two very recent concurrent works (Zhang et al., 2023) and (Pang et al., 2023) have initiated the use of such metric learning approaches to approximate geodesic distances on meshed surfaces for graphics and vision applications. In particular, (Zhang et al., 2023) introduces a general framework,

neural geodesic field (NeuroGF), consisting of an initial Siamese *embedding module* $\phi$ mapping input points to a latent space, combined with a final multilayer perceptron (MLP) $\rho$ as the *distance approximation module*. See Figure 1. (Pang et al., 2023) suggests a similar network architecture for approximating SPD on mesh surfaces but incorporates a hierarchical pooling module within their suggested architecture. Both (Zhang et al., 2023) and (Pang et al., 2023) are state-of-the-art (SOTA) neural approaches to approximating SP-distances for smooth meshed surfaces. However, as we will see later, these models (especially (Pang et al., 2023)) do not yet scale to the size encountered in terrain graphs.

**Our model.   Terrains.** Let $\Omega \subset \mathbb{R}^2$ be a triangulation of a bounded polygonal region. With a slight abuse of notation, we also denote this region by $\Omega$. Let $V$ be the set of vertices of $\Omega$; set $n = |V|$. Let $h : \Omega \to \mathbb{R}$ be a height function. We assume that the restriction of $h$ to each triangle of $\Omega$ is a linear map. Given $\Omega$ and $h$, the graph of $h$, called a *(polygonal) terrain* and denoted by $\Sigma = (\Omega, h)$, is an $xy$-monotone triangulated surface whose triangulation is induced by $\Omega$. For any point $v = (x, y) \in \mathbb{R}^2$, we define its *lift* as the point $\hat{v} = (x, y, h(x, y)) \in \mathbb{R}^3$.

**Paths and terrain graphs.** Our focus is on processing high-resolution terrain data sets, in which triangles in $\Omega$ will be tiny. Therefore we ignore the interiors of triangles and consider the terrain as a weighted graph $G_h = (V, E)$, where $V$ and $E$ are the vertices and edges of $\Omega$, and the weight of an edge $(u, v)$ is $\|\hat{u} - \hat{v}\|$ (the Euclidean distance between the lifts of $u$ and $v$). We refer to $G_h$ as a *terrain graph*. [1] For two vertices $\hat{u}, \hat{v}$, we assume that a path on $\Sigma$ between $\hat{u}$ and $\hat{v}$ lies along the edges of $G_h$, and thus the shortest path

---

[1]For example, most of the terrain data is available as grid DEM, i.e., height function over points on a regular grid, in which case each grid point is connected to its eight neighbors; see e.g. (Arge et al., 2001).

between $\hat{u}$ and $\hat{v}$ on $\Sigma$ is represented as the shortest path between them in the graph $G_h$. This simplified assumption is justified because of high-resolution data and because we are interested in computing the geodesic distance approximately. Furthermore, most approximation algorithms also construct a discrete graph and compute shortest paths in the resulting graph. This assumption enables us to simplify the problem at a small loss in accuracy and to use graph neural network architectures (Scarselli et al., 2009).

**Our contributions.** This paper aims to develop an efficient and compact neural data structure for the estimation of shortest path distance (SPD) for massive terrain graphs at an unprecedented scale. While (Pang et al., 2023) shows promising results for SPD estimation on well-behaved triangular meshes of smooth 3D geometric shapes (e.g. from ShapeNet), terrain graphs pose a unique challenge as they are significantly larger in size. We instead explore the use of the conceptually simpler NeuroGF framework proposed in (Zhang et al., 2023) to develop an effective model for massive terrain graphs.

Our main contributions are as follows: We propose a simple *decoupled-NeuroGF* framework for answering shortest path distance queries approximately on massive terrain graphs, which has a two stage training process. In the first stage, only the Siamese embedding module $\phi$ is trained to approximate the SP-distance via the $L_1$ distance in the latent space, and its weights are subsequently frozen. In the second stage, we finetune the final MLP (which refines the distance approximation) using the fixed embeddings output by the Siamese network. We call this MLP part the *distance-computation module*. This decoupling of the two stages of NeuroGF has several advantages:

- **Better performance.** As we show in Section 4, it achieves superior performance in practice, with up to a 10% reduction in relative error as compared to the end-to-end NeuroGF pipeline.

- **Efficient mixed-training strategies.** Training the Siamese embedding module $\phi$ takes significantly longer than training the distance-computation MLP $\rho$. As the size of terrain grows, training $\phi$ becomes very expensive. Our decoupled-NeuroGF uses a mixed training-strategy: We first train the embedding module $\phi$ using only a *coarsened terrain graph*. While keeping the coarse embedding module $\phi$ frozen, we then train the distance-computation module $\rho$ to improve and refine distance estimates using the *original full-resolution terrain graph*. This strategy allows us to scale our neural model to very large terrains.

- **Efficient updates for terrain changes.** Unlike 3D mesh graphs, terrains are inherently dynamic as they are subject to uncertainty (due to measurement errors) and continuous changes driven by natural processes. Re-training the entire NeuroGF model whenever the height function is modified is expensive. With our de-coupled training approach, the previously trained $\phi$ can be re-used to map nodes to an initial representation and we re-train only the final distance-computation module $\rho$ using the perturbed graph with the updated height function. Via our de-coupled training strategy, we can efficiently update our neural DS whenever the height function changes and maintain high quality SPD approximations.

In Section 4, we demonstrate the utility of our decoupled-NeuroGF and its mixed training strategy for large terrain graphs at an unprecedented scale. Specifically, we evaluate real-world terrains containing up to 16 million nodes, a significant increase compared to prior algorithmic approaches. We can train a small model with around 70000 parameters on a terrain DEM with 16 million nodes in a matter of hours in contrast to the previous best-known (non-neural) algorithms which require several days of pre-processing time for terrains with 180,000 nodes. Our mixed training strategy also leads to orders of magnitude speedup in training, while still outperforming previous SOTA neural based approaches. Our experiments also show that our decoupled-NeuroGF supports efficient updates (retraining of only the distance-computation MLP module) as the input terrain graph undergoes small changes.

## 2. Preliminaries

Given a height function $h : \Omega \to \mathbb{R}$, our input is a terrain graph $G_h = (V, E; X_h, w)$ where $X_h = \{(x, y, h(x, y)) : v = (x, y) \in V\}$ (the set of lifts of nodes in $V$ w.r.t. the height function $h : V \to \mathbb{R}$) and $w : E \to \mathbb{R}$ assigns edge weights. We often omit the subscript $h$ from the notations when its choice is clear. In practice, terrain data is often given as a grid of evenly spaced measurements within a constrained region. In this case, we simply set $V$ to be the set of grid vertices, and for each node (grid point) $v$, we connect it to each of its 8 neighbors around it. Our goal is to estimate the shortest path distance $d_{G_h}(u, v)$ w.r.t. the terrain graph $G_h$.

**Graph neural networks.** Graph neural networks (GNNs) are a class of deep learning model specifically designed to work with graph structured input (Scarselli et al., 2009). In practice, the most commonly used graph neural network architecture is the *message passing* neural network (Gilmer et al., 2017; Jegelka, 2022) where node features are learned by aggregating neighboring node features. Formally, suppose we are given a graph $G = (V, E, X)$, where $X = \{x_v \in \mathbb{R}^D : v \in V\}$ represents the *initial node features*; that is, $x_v^{(0)} := x_v$. Then at each $\ell$ of the $L$ layers of

the GNN, $\ell \in [1, L]$, we update the node features at each node as follows:

$$x_v^{(\ell)} = f_{\text{up}}\left(x_v^{(\ell-1)}, f_{\text{agg}}\left(\{x_u^{(\ell-1)} : u \in \mathcal{N}(v)\}\right)\right), \quad (1)$$

where $x_v^{(\ell)}$ denotes the updated feature of node $v \in V$ at layer $\ell$, $\mathcal{N}(v)$ is the set of neighbors of $v$, $f_{\text{agg}}$ aggregates previous features from the neighbors (including edge weights) in a permutation invariant manner, and $f_{\text{up}}$ computes the updated node feature from the aggregated information. In practice, graph convolutional networks (GCN) (Kipf & Welling, 2016) and graph attention networks (GAT) (Veličković et al., 2018) are among the most commonly used variants of GNNs and have been explored for learning algorithmic tasks (Veličković et al., 2022; Veličković et al., 2020). We consider both GCNs and GATs in our exploration of the design space of NeuroGF. See Appendix A for exposition regarding GCNs and GATs. Once trained, a GNN can be applied to any graph with any combinatorial structure because the learnable parameters in the aggregate and update functions are shared across all nodes. A single forward pass through the GNN takes time $O(|V| + |E|)$, linear with respect to the number of nodes and edges.

In this paper, we also use the transformer model, which captures global relations between nodes via self-attention but is computationally expensive ($O(|V|^2)$). We leave the description of transformers in Appendix B.

## 3. Neural Shortest Path Data Structures

### 3.1. The NeuroGF pipeline (Zhang et al., 2023)

We begin by reviewing the framework of NeuroGF (Zhang et al., 2023). Given a metric space $(\mathcal{X}, d_\mathcal{X})$, the goal of the NeuroGF framework is to approximate the target geodesic (or SP-distance in our case) function $f = d_X : \mathcal{X} \times \mathcal{X} \to \mathbb{R}$. That is, we approximte $d_\mathcal{X}(x_1, x_2)$ by $\rho(\phi(x_1), \phi(x_2))$, where $\phi : \mathcal{X} \to \mathbb{R}^D$ and $\rho : \mathbb{R}^D \to \mathbb{R}$. See Figure 1 for an illustration. To motivate this formulation, consider a traditional metric learning scenario where $\rho$ simply represents an $L_p$ distance in the embedding space. In this case, NeuroGF is simply a Siamese network that uses $\phi$ as an embedding module to map input node features to a high-dimensional latent space $Z = \mathbb{R}^D$. However, by using an MLP in the place of $\rho$, we can approximate distance functions that are not $L_p$ embeddable. Thus, the entire NeuroGF framework is intuitively an initial *Siamese embedding module* $\phi$ followed by a final *distance computation module* $\rho$.

Recall that we represent the terrain as a graph $G_h = (V, E, X, w : E \to \mathbb{R}^+)$ where $w$ represents the edge weights and $X$ is the set of input node features, $\hat{v} = (x, y, h(x, y)) \in \mathbb{R}^3$. Our training set $S$ is made up of a subset of pairs of vertices and their shortest path distances. The final loss function is:

$$\mathcal{L}(S, G_h, \phi, \rho)$$
$$= \frac{1}{|S|} \sum_{(u, v, d_{G_h}(u,v)) \in S} \left(d_{G_h}(u, v) - \rho(\phi(\hat{u}), \phi(\hat{v}))\right)^2 \quad (2)$$

### 3.2. Instantiation of the NeuroGF

In this paper, we instantiate the embedding module $\phi$ with several choices of neural network architectures: (1) a standard MLP (2) GCN (3) GAT and (4) transformer architecture. The specific input for $\phi$ varies depending on the chosen architecture. The instantiation of $\rho$ will be discussed at the end of this subsection. We describe the MLP and GNN instantiation of $\phi$, denoted as MLP and GNN respectively. We leave the description of the transfomer to Appendix B.2.

**Multilayer perceptron (MLP).** The simplest approach for instantiating the embedding module $\phi$ is to use an MLP. Given a pair of nodes $u, v$ and their initial node features $x_u = \hat{u}, x_v = \hat{v}$, we train a neural model to predict $\mathcal{F}_{\text{MLP}}(x_u, x_v) \approx \rho(\text{MLP}(x_u), \text{MLP}(x_v))$. Note that $\mathcal{F}_{\text{MLP}}$ does not utilize the input graph structure and depends solely on the input feature representations of individual nodes. It aims to directly learn the shortest path distance function $d_G : V \times V \to \mathbb{R}$ from the train set as a general function approximation problem. However, our $d_{G_h}$ is not an arbitrary function – it is induced by $G_h$. Thus, using MLPs, as opposed to GNNs or transformers, as $\phi$ to process nodes independently fails to capture relational information from the terrain graph $G_h$.

**Graph neural network.** In general, when given a terrain graph $G$ and a node $v$ and when we are using a GNN as the initial embedding module, we write the output embedding $\phi(v; G) = \text{GNN}_G(v)$ to emphasize the incorporation of graph structure as the input of $\phi$. Hence the full model now is $\mathcal{F}_{\text{GNN}}(x_u, x_v; G) = \rho(\text{GNN}_G(x_u), \text{GNN}_G(x_v))$. We use GCNs and GATs as the choices for the (message passing) graph neural network $\text{GNN}_G$: GCNs are the specific message passing architecture originally used in (Zhang et al., 2023); we include GATs (Veličković et al., 2018) as they are able to attend to more 'important' vertices.

For a general GNN, the output node embeddings are computed as described in Equation (1). In our setup, for each node $v \in V$, the initial input node features are $h_v^{(0)} = \hat{v} \in \mathbb{R}^3$, and the output of $\phi(\hat{v})$ is the final node embedding $h_v^{(L)}$, where $L$ represents the number of layers in the GNN. In practice, we also incorporate the terrain edge weights in the computation of the next node features (see details in Appendix A).

Each forward pass of the GNN is more than one forward pass of the MLP, as it takes time $O(|V|+|E|)+O(|S_{\text{train}}|)$, while in the case of MLP, a forward pass only linear in the number of training set size. However, one would expect

that the embedding learned by the GNN is much better than that of MLP as with even a small number of training pairs, the utilization of the graph structure allows the embedding module to be aware of global relation between graph nodes. Finally, while the GNN can incorporate the graph structure in each node embedding output, it can only see within an $L$-hop neighborhood of each node where $L$ is the number of layers in the GNN.

**Instantiation of the distance computation module.** We explore two choices of the distance-computation module $\rho$: (1) $\rho$ is an $L_p$ distance function or (2) parameterize $\rho$ as an MLP. The case that $\rho$ is an $L_p$ distance represents the traditional metric-learning paradigm, where the embedding module $\phi$ is a Siamese network. In the case where we parameterize $\rho$ as a final MLP (as proposed by NeuroGF (Zhang et al., 2023)), the model more accurately approximates distance functions that may not be isometrically embeddable into $L_p$ space. Given some instantiation of $\phi$, the final output is $\mathsf{MLP}([\phi(v_i)+\phi(v_j), |\phi(v_i)-\phi(v_j)|])$, where we concatenate $\phi(v_i) + \phi(v_j)$ and the absolute difference $|\phi(v_i) - \phi(v_j)|$.

### 3.3. De-coupled training process

We now describe our new *de-coupled training process* for the NeuroGF framework which leverages the distinct roles of the embedding module $\phi$ and the distance computation module $\rho$; $\phi$ usually does not capture global information in a highly accurate manner. For instance, if $\phi = \mathsf{GNN}$, the receptive field is limited by the number of layers in the GNN and restricts the $\phi$ from capturing the global terrain structure. The distance-decoding stage, when implemented via the MLP, addresses these limitations by learning how to better combine the embeddings to produce a more accurate approximation of the target distance. Therefore, we propose a two-stage training process which explicitly de-couples the process of embedding generation via $\phi$ and distance computation via $\rho$. This decoupling of embedding generation and distance computation is a novel perspective that has so far not been explored in prior work and offers significant practical advantages (which we will explore in Section 4). Our new training process proceeds as follows:

- **Stage 1: Training the embedding module.** We train only the embedding module $\phi$ on training set $S_{\mathrm{train},1}$ as a Siamese network to generate node embeddings and use $L_1$ distance between embeddings $\|\phi(u) - \phi(v)\|_1$ as the estimate of $d_G(u, v)$ to compute training loss.

- **Stage 2: Training the distance-computation module.** After training $\phi$, we *freeze its weights* and use its output embeddings as input to train the final distance computation module $\rho = \mathsf{MLP}$ using a second training set $S_{\mathrm{train},2}$.

Intuitively, this de-coupled training strategy uses $\phi$ to generate a 'decent' first embedding and delegates fine-tuning to the final distance computation module $\rho = \mathsf{MLP}$ to correct any errors incurred by $\phi$. De-coupling the training of $\phi$ and $\rho$ also allows us to quickly adapt to any changes to the input terrain as we can avoid re-training $\phi$ and simply re-train the final $\rho$, which is faster than re-training the entire NeuroGF.

**Mixed coarse-to-refined de-coupled training.** This de-coupled training process can be extended to a *mixed coarse-to-refined training framework* (M-CTR) to further reduce training costs, especially when the initial embedding module ($\phi$) is implemented using computationally expensive architectures like GNNs or transformers. In particular, each forward pass in stage 1 takes $O(|V| + |E| + |S_{\mathrm{train},1}|)$ time if we use GNN for $\phi$, or $O(|V|^2 + |S_{\mathrm{train},1}|)$ if we use transformer. This step is the computational bottleneck, and takes orders of magnitude longer time than just training the MLP in the distance computation module (which takes only $O(|S_{\mathrm{train},2}|)$ for each forward pass).

The key idea of M-CTR training is to leverage a coarsened version of the input terrain to train the initial embedding module $\phi$ in order to reduce the computational bottleneck. One can compute the downsampled (coarsened) terrain graph via many strategies. In our experiments, since our terrain graph $G_h = (V, E, X, w)$ is induced by a $N_1 \times N_2$ grid, we simply produce coarsened graph by downsample the grid to be $\frac{N_1}{k} \times \frac{N_1}{k}$ for some integer $k$. Denote the induced downsampled terrain graph by $G'_h$.

Instead of training $\phi$ on the full graph $G_h$, we first train $\phi$ on this coarsened version $G'_h$ using a training set $S_{\mathrm{train},1}$ constructed from $G'_h$. Then, we freeze $\phi$ and train only the distance computation module $\rho$ using a new training set $S_{\mathrm{train},2}$ from the original high-resolution terrain graph $G_h$. Note that the number of training samples in $S_{\mathrm{train},2}$ can be much larger than in $S_{\mathrm{train},1}$. This distance computation module $\rho$ learns to correct inaccuracies arising from the initial coarse embedding trained on coarsened graph $G'_h$.

## 4. Experiments

We first explore the design space of the NeuroGF pipeline (Zhang et al., 2023), both in terms of the choice of neural networks for the embedding module $\phi$, and the effect of the distance computation MLP module $\rho$. We then show the benefit of our decoupled-NeuroGF framework and our mixed-training strategy (M-CTR) as compared to our implementation of the original NeuroGF framework for real terrains. We also compare our method to GeGNN, another SOTA neural data structure (Pang et al., 2023). We measure two error metrics: (1) the average relative error with respect to the true SPD (2) accuracy, defined as the percentage of test instances where relative error is below 2%. As we will

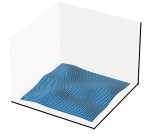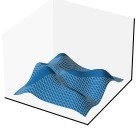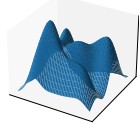

*Figure 2.* Synthetic terrain surfaces of three of the synthetic terrains (Gaussian amplitudes visualized are $a \in \{1.0, 4.0, 10.0\}$).

| $a$ | $k_X$ | De-coupled | GAT + $L_1$ |
|---|---|---|---|
| 1.0 | 4.1 | **0.52 ± 0.61** | 0.61 ± 0.88 |
| 4.0 | 4.4 | **1.32 ± 1.84** | 1.66 ± 3.19 |
| 10.0 | 4.7 | **1.86 ± 3.48** | 3.29 ± 6.76 |

*Table 1.* Approximate doubling dimension (denoted by $k_X$) compared again the average relative error (%, ↓) for the Siamese model and the de-coupled training procedure on synthetic terrains. We also include the maximum amplitude of the Gaussians in the synthetic terrain and denote it as $a$. Notice that the de-coupled training procedure can help mitigate the errors introduced by the Siamese embedding.

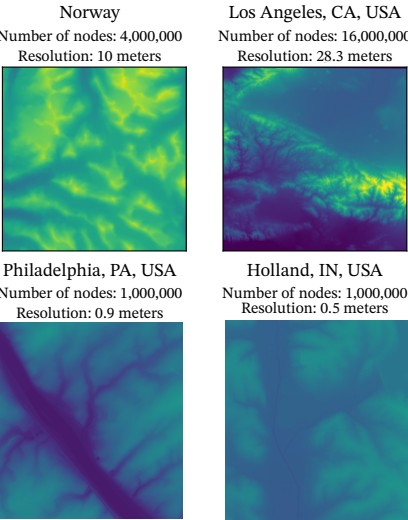

*Figure 3.* Visualization of each terrain's height function. Lighter areas correspond to more mountainous regions.

see later, our new strategies significantly improve previous neural approaches on large scale terrains, e.g, an improvement (reduction) of more than 10% of error (Table 3) as well as even more significant improvements for weighted terrains (Table 4) and SPD estimation in for terrains with uncertainty (Table 5). Details regarding training, hyperparameters, etc. are given in the Appendix C.

### 4.1. Experimental setup

We use two types of data: synthetic terrain DEMs of different "complexity", and real terrain DEMs with up to 16 million nodes.

**Synthetic terrains.** To understand the performance of different model designs, we generate a series of terrains with the same size but different complexity. These synthetic terrain graphs are generated using a mixture of 2D Gaussians over $[0, 10]^2$ and the amplitudes from $\{1.0, 2.0, 4.0, \dots, 18.0\}$. See Figure 2 for examples. Each terrain graph is induced by a $50 \times 50$ grid and has 2500 nodes. To generate the downsampled terrain graphs for M-CTR training, we downsample the grid to a $25 \times 25$ grid (625 nodes).

**Real terrains.** We use four real-world terrain DEMs: (1) Troms region of Norway (4M nodes) (2) Los Angeles (LA),

California, USA (16M nodes) (3) Holland, Indiana, USA (1M nodes), and (4) Philadelphia (Phil.), Pennsylvania, USA (1M nodes). See Figure 3 for illustrations. We point out the unique complexities of the LA and Norway terrains: the LA dataset is the largest, with 16 million nodes, while the Norway dataset represents a highly mountainous region.

**Design space and experimental setup.** The framework of NeuroGF (Zhang et al., 2023) consists of an embedding module $\phi$ and a distance-computation module $\rho$. We instantiate the embedding module $\phi$ by (1) a MLP, (2) GCN (Kipf & Welling, 2016), (3) GAT (Veličković et al., 2018), and (4) a transformer (Vaswani et al., 2017). In what follows, we use X + Y, X $\in$ { MLP, GCN, GAT, Transformer } and Y $\in \{L_p,$ MLP $\}$ to represent all these model designs. The *original NeuroGF* of (Zhang et al., 2023) corresponds to the setup X + MLP with $X \in \{$MLP, GCN, Transformer$\}$. For our *de-coupled NeuroGF* (indicated by 'de-coupled'), we first train GAT + $L_1$, freeze its weights, and then train an MLP for the distance-computation module.

### 4.2. Design space of NeuroGF and de-coupled NeuroGF

**Design space of NeuroGF.** First, we test the relative error w.r.t. SP-distances by different model designs (including GeGNN of (Pang et al., 2023)) for synthetic terrains. We leave a detailed figure and results in Appendix C.3, and only list the key observations here: (1) GCN and GAT far outperform MLP or transformer based instantiation of the embedding module $\phi$ (cf. Appendix C.3, Figure 5) . (2) GeGNN (Pang et al., 2023) performs much worse than the original NeuroGF (Zhang et al., 2023) with any instantiation of the embedding module (X+MLP in our setup). (3) Using only X+$L_1$ consistently outperforms X+MLP (the original NeuroGF). Interestingly, using $L_1$ in the latent space outperforms using $L_2$ in the latent space as well; that is, X+$L_1$ is (sometimes much) better than X+$L_2$ for any choice of X (cf. Appendix C.3, Figure 5). From these observations, we narrow our architectural explorations and exclude the MLP and Transformer architectures, as well as X+$L_2$ from our further experiments.

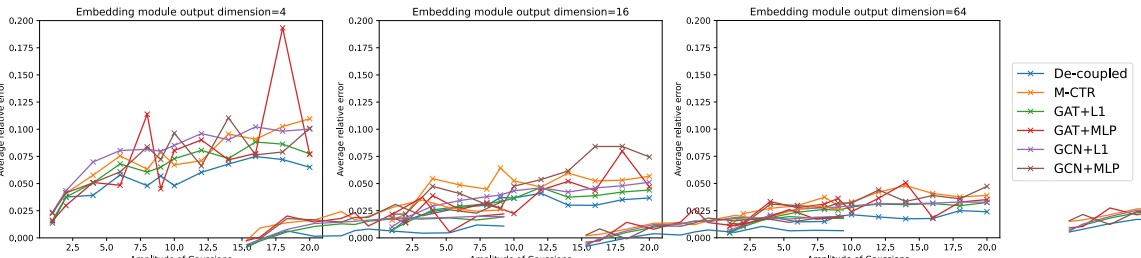

*Figure 4.* Average relative error (y-axis) of each GNN based model against the the amplitude of the Gaussians ('complexity') in an synthetic terrain. Each plot corresponds to a different latent space dimension for the embedding module $\phi$.

| Model | Relative Error ($\%, \downarrow$) | Accuracy ($\%, \uparrow$) | Training Time (min, $\downarrow$) |
|---|---|---|---|
| GAT+$L_1$ | $0.92 \pm 0.95$ | 91.5 | 153 |
| GAT+MLP | $1.32 \pm 2.03$ | 89.4 | 155 |
| GeGNN | $13.10 \pm 2.10$ | 34.5 | **127** |
| Decoupled (ours) | $\mathbf{0.84 \pm 1.66}$ | **92.3** | 200 |

*Table 2.* Results for a downsampled 250x250 version of Norway-250. The GAT+MLP is the same as the NeuroGF framework. Note that the training time for the de-coupled version of NeuroGF includes both the time for training GAT+$L_1$ as well as the time to finetune the final MLP distance computation module (which takes approximately 1hr). Best results are highlighted in red.

**Effectiveness of de-coupled NeuroGF.** In Figure 4, we compare our de-coupled NeuroGF framework with the original NeuroGF with a GNN instantiation (i.e, GAT+MLP and GCN+MLP), as well as with two baseline Siamese architectures, GAT+$L_1$ and GCN+$L_1$, over a series of synthetic terrains. We include a comparison with the mixed coarse-to-refined training approach, M-CTR($625 \rightarrow 2500$), where we train the initial embedding module $\phi$ on a coarse version of each synthetic terrain with $625$ nodes, then freeze $\phi$, and train only the distance-computation module $\rho$ on the refined version of the terrain with $2500$ nodes. We plot the average relative error in $y$-axis against the 'complexity' (the maximum amplitutude of Gaussians) of terrains in $x$-axis.

As we can see in Figure 4, our de-coupled NeuroGF framework consistently outperforms both the original NeuroGF and the Siamese baselines. We note that our mixed-training strategy M-CTR performs comparable or better than the two NeuroGF instantiations (GAT+MLP and GCN+MLP), even though its embedding module is trained on a much coarser terrain graph (and thus faster). This improvement is consistently observed across all output embedding sizes, further emphasizing the robustness of the de-coupled NeuroGF approach. Finally, using GAT as the initial embedding module outperforms GCN, so we will use GAT as the initial embedding module for real terrain experiments below. An expanded version of Figure 4 for a larger set of embedding sizes is in Appendix C.

**Effect of terrain complexity.** In Figure 4, we note that as the maximum amplitude of the Gaussians in the synthetic terrain increases, the performance gap between the

de-coupled framework and the Siamese baseline becomes even more pronounced. The de-coupled training of the MLP-based distance computation module proves crucial in synthetic terrains where the shortest path contour deviates substantially from the Euclidean ($L_2$) distance contour.

Overall, we believe that the approximation quality of each neural method depends on the intrinsic "complexity" of the terrain graph metric, which we quantify as the *doubling dimension* of the terrain (denoted by $k_X$). A metric space $X$ has doubling constant $k$ if for all $r > 0$, every $r$-ball (radius $r$-neighborhood around any point in $X$) can be covered by at most $k$ $\frac{r}{2}$-balls and and the *doubling dimension* of $X$ is the base-2 logarithm of the doubling constant ($k_X = \log_2(k)$). The relationship between doubling dimension and approximation quality is theoretically motivated by Theorem 2.1 of (Naor & Neiman, 2012), which states that every metric space can be approximately embedded into Euclidean space with distortion dependent on the doubling dimension of the space. In Table 1, we compare the approximate doubling dimension against the relative error of GAT + $L_1$ as well as our de-coupled neural method. We observe that as doubling dimension increases, the relative error incurred by the Siamese embedding approach GAT+$L_1$ also increases. Additionally, our de-coupled training approach seems to help adjust the errors from the Siamese embedding approach. The relative error of the de-coupled approach also seems to increase at a lower rate than that of GAT+$L_1$.

| Dataset | Model | Relative Error (%, ↓) | Accuracy (%, ↑) | Training Time (min, ↓) |
|---|---|---|---|---|
| Norway (4M nodes) | Full-GAT+$L_1$ | $0.94 \pm 0.95$ | 90.5 | 683 |
| | Coarse-GAT+$L_1$ | $1.05 \pm 0.99$ | 89.2 | **150** |
| | M-CTR (ours) | $\mathbf{0.90 \pm 1.49}$ | **91.1** | 183 |
| | Our improvement | **-14.2 %** | **+1.9%** | - |
| LA (16M nodes) | Full-GAT+$L_1$ | - | - | 2733 |
| | Coarse-GAT+$L_1$ | $1.04 \pm 1.09$ | 85.51 | **137** |
| | M-CTR (ours) | $\mathbf{0.88 \pm 3.47}$ | **94.41** | 183 |
| | Our improvement | **-15.4 %** | **+8.9%** | - |
| Philadelphia (1M nodes) | Full-GAT+$L_1$ | $\mathbf{0.21 \pm 0.29}$ | **99.6** | 147 |
| | Coarse-GAT+$L_1$ | $2.07 \pm 1.47$ | 30.1 | **31** |
| | M-CTR (ours) | $0.51 \pm 0.71$ | 94.4 | 83 |
| | Our improvement | **-75.4%** | **+64.3%** | - |
| Holland (1M nodes) | Full-GAT+$L_1$ | $\mathbf{0.11 \pm 0.12}$ | **99.9** | 123 |
| | Coarse-GAT+$L_1$ | $2.06 \pm 1.54$ | 65.1 | **32** |
| | M-CTR (ours) | $0.86 \pm 2.29$ | 90.8 | 95 |
| | Our improvement | **-58.3%** | **+25.7 %** | - |

*Table 3.* Results for LA (16M nodes), Norway (4M nodes), Philadelphia (1M nodes), and Holland (1M nodes). Our M-CTR and Coarse-GAT+$L_1$ have similar training time, however, our M-CTR has much better performance (best performance marked in red). In the last row for each dataset we list our improvement (of M-CTR) over Coarse-GAT+$L_1$): note that for relative error, we reduce error (thus the '-' sign), while for accuracy, we improve (thus the '+' sign). While the full GAT + $L_1$ has better performance for Philadelphia and Holland than our M-CTR, we note that it takes much longer to train. For both Philadelphia and Holland, our M-CTR model has much better performance than Coarse-GAT+$L_1$. For Norway, our model has slightly better performance than Full-GAT+$L_1$, but is much faster. We extrapolate the training time for Full-GAT +$L_1$ on LA (which did not finish after 30+ hours) from the training time for Norway under the assumption that training time is linear to input graph size.

## 4.3. Decoupled-NeuroGF on massive real terrains

For all terrains, the time NeuroGF (and all variants, including the Siamese network) needs to complete a SPD query at inference time is at most 10 ms. See Appendix C for a table of the time needed to compute embeddings. We use a downsampled 250x250 version of Norway as a case study to demonstrate the utility of the de-coupled NeuroGF training procedure compared to the entire end-to-end NeuroGF pipeline, the Siamese network approach, and the other SOTA model GeGNN. The downsampled graph is used because GeGNN cannot be trained on terrain graphs exceeding 62,500 nodes due to memory limitations. See Table 2 for our results. Similar to what we observed for synthetic terrains, we see that our de-coupled approach has the best performance as compared to all tested baselines. Interestingly, GAT+$L_1$ outperforms both GeGNN and NeuroGF (which is denoted as GAT+MLP).

We evaluate our de-coupled training pipeline, in particular, our mixed coarse-to-refined strategy M-CTR, to full resolution Los Angeles (16M nodes), Norway (4M nodes), Holland (1M nodes), and Philadelphia (1M nodes). In particular, terrains at the size of Los Angeles and Norway have not previously been explored. For our M-CTR, on all terrains, we first train the embedding network $\phi$ on a (coarse) downsampled version of each terrain (250x250 for Norway and LA, 50x50 for Philadelphia and Holland. We then fine-tune the final distance computation MLP $\rho$ on train instances over the full-resolution terrain. We compare our M-CTR

with two approaches: (a) the GAT+$L_1$ on the full-resolution terrain, represented by Full-GAT+$L_1$ in Table 3; and (b) the Coarse-GAT+$L_1$, where we simply apply the coarse embedding network $\phi$ trained over the same downsampled version, then use $L_1$ distance in the latent space.

As seen in Table 3, our M-CTR has much better relative error and accuracy than Coarse-GAT+$L_1$, while the training times are very similar. In particular, the improvements for the LA, Philadelphia, and Holland are notable for both error and accuracy – about 10% for both on LA and over 20% for both on Philadelphia and Holland. For Philadelphia and Holland, the Full-GAT+$L_1$ has better performance than M-CTR but our M-CTR is much faster. Interestingly, our M-CTR has better performance than the Full-GAT+$L_1$ on Norway as well while it is much faster (only takes a quarter of the time needed by the full model). The training of Full-GAT+$L_1$ on the 16M LA model is prohibitively expensive in both time and memory, taking 30+ hours without completion. Note that if the training time scales linearly, the training time for Full-GAT+$L_1$ is expected to be approximately four times that of training Full-GAT+$L_1$ on the Norway dataset.

**Weighted terrains.** The improvement of our M-CTR is even more significant when the input terrain graph is weighted. Following usual practices in GIS, we assign each edge $(u, v)$ by $(1 + \theta)\|\tilde{u} - \tilde{v}\|_2$ where $\theta$ is the absolute value of the angle of elevation. In Table 4, we compare our M-CTR with the Coarse-GAT+$L_1$. Note that our improvement for the larger LA dataset is significant: the relative

| Datasets | Models | Rel. Err. (%, ↓) | Acc. (%, ↑) | | Datasets | Models | Rel. Err. (%, ↓) | Acc. (%, ↑) |
|---|---|---|---|---|---|---|---|---|
| Norway | Coarse-GAT+$L_1$ | $2.30 \pm 2.96$ | 62.14 | | Norway | GAT + $L_1$ | $19.4 \pm 3.04$ | 0.078 |
| | M-CTR (ours) | $\mathbf{2.04 \pm 3.53}$ | **68.06** | | | Decoupled (ours) | $\mathbf{1.76 \pm 2.30}$ | **70.0** |
| | Our improvement | **-11.3%** | **+5.9%** | | | Our improvement | **-90.9%** | **+69.9%** |
| LA | Coarse-GAT+$L_1$ | $3.55 \pm 2.76$ | 27.53 | | LA | GAT + $L_1$ | $2.27 \pm 2.06$ | 34.6 |
| | M-CTR (ours) | $\mathbf{1.91 \pm 4.66}$ | **69.42** | | | Decoupled (ours) | $\mathbf{0.77 \pm 1.04}$ | **94.1** |
| | Our improvement | **-46.2%** | **+41.89%** | | | Our improvement | **-66.0%** | **+59.5%** |
| Phil. | Coarse-GAT+$L_1$ | $2.43 \pm 4.62$ | 61.4 | | Phil. | GAT + $L_1$ | $29.8 \pm 6.84$ | 0.048 |
| | M-CTR (ours) | $\mathbf{1.32 \pm 2.96}$ | **83.9** | | | Decoupled (ours) | $\mathbf{6.14 \pm 7.88}$ | **26.4** |
| | Our improvement | **-45.7%** | **+22.5%** | | | Our improvement | **-79.4%** | **+26.3%** |
| Holland | Coarse-GAT+$L_1$ | $2.48 \pm 6.20$ | 61.9 | | Holland | GAT + $L_1$ | $48.79 \pm 7.08$ | 0.092 |
| | M-CTR (ours) | $\mathbf{0.82 \pm 1.69}$ | **93.3** | | | Decoupled (ours) | $\mathbf{9.35 \pm 13.2}$ | **17.3** |
| | Our improvement | **-66.9%** | **+31.4%** | | | Our improvement | **-81.0%** | **+17.2%** |

*Table 4.* Weighted terrains results. The improvement has '-' sign for error as we reduce it and '+' for accuracy as we increase it. We note that our M-CTR approach far out-performs the Coarse-GAT+$L_1$ approach for all weighted terrains.

*Table 5.* Results for uncertainty models. Improvement has '-' sign for error as we reduce it and '+' for accuracy as we increase it. For each terrain, our de-coupled approach outperforms GAT+$L_1$.

error for Coarse-GAT+$L_1$ almost doubles that of our M-CTR, while its accuracy is only half of ours.

### 4.4. Dynamic terrain changes

Our de-coupled pipeline allows us to efficiently handle updates to the terrain. We illustrate this advantage when considering the presence of *uncertainty* in terrain measurements. See Appendix C for the improvement by our approaches for a synthetic scenario where there are *edge-weight updates* to the terrain.

DEMs can have inherent uncertainties from measurement errors or incomplete data (Banerjee et al., 2003; Zhang et al., 2015). For example, (Zhang et al., 2015) treats terrain height as a random variable within a range (instead of a fixed value): the height of any vertex $v$ lies in $[h(v) - 0.05, h(v) + 0.05]$ for a height function $h : \Omega \to \mathbb{R}$. Then one can sample multiple potential terrain height maps, and estimate SPD as the average shortest path distance across these multiple samples. Specifically, we randomly sample $K$ terrain instantiations, $\{G_{h_1}, \ldots, G_{h_K}\}$, each with unique shortest path distances. The ground truth shortest path between vertices $u$ and $v$ is computed as $\frac{1}{K} \sum_{i=1}^{K} \mathrm{d}_{G_{h_i}}(u, v)$.

If we use a neural data structure to estimate the SPD, in general, one would need to *re-train $K$ models*, one for each terrain instantiation $G_{h_i}$. However, with our decoupled training procedure, we only need to train an embedding map *once*, and retrain only the much cheaper distance computation MLPs $K$ times. This improves the time needed to estimate average SPDs by a factor proportional to $K$. To test, we carry out this model on downsampled Holland, Philadelphia, Norway and LA terrains (250x250 for Norway and LA, 50x50 for Holland and Philadelphia). For $K = 50$, our entire retraining takes around 12 hours, while retraining GAT+$L_1$ 50 times would take 100 hours. In Table 5, we compare the SPD estimation accuracy of the estimate

by averaging the output of our retrained decoupled models (indicated as 'Decoupled (ours)' in the table) and the estimate of a single GAT+$L_1$ (which has comparable training time as ours). Notice that Holland and Philadelphia have much larger errors as compared to Norway and LA, possibly because a $\pm 0.05$ perturbation in vertex height represents a far more significant relative change on those comparatively flatter terrains. Overall, our approach achieves dramatically better performance with comparable training time.

## 5. Conclusion

We demonstrate the effectiveness of neural data structures in approximating SPD on large-scale terrain graphs. By leveraging learned embeddings and efficient distance-computation modules, our de-coupled NeuroGF is scalable and allows for more efficient updates with dynamic terrain changes. Our decoupled NeuroGF offers a promising alternative to traditional data structure for SPD queries for massive terrains. This opens more venues for exploring *neural data structures* that can summarize input data in a succinct manner yet support efficient queries.

## Impact statement

This paper presents work whose goal is to advance the field of Machine Learning. There are many potential societal consequences of our work, none which we feel must be specifically highlighted here.

## Acknowledgements

Work by Pankaj K. Agarwal was partially supported by NSF grants CCF-20-07556, CCF-22-23870, and IIS-24-02823, and by the Binational Science Foundation Grant 2022131. Work by Samantha Chen and Yusu Wang is additionally supported by NSF grants CCF-2112665 and CCF-2310411.

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

## A. Graph Neural Networks

We provide exposition regarding different message passing GNN architectures architectures. Recall that the general form of the a message passing GNN was given in Equation (1). Most popular architectures, such as GCN or GAT, follow the general format in Equation (1) with differences between architectures arising from the implementation of the aggregation and update functions. A GCN uses normalized mean pooling as the update and aggregation function so at each layer $\ell$ the node is updated as

$$x_v^{(\ell)} = \sigma \left( \sum_{u \in \mathcal{N}(v) \cup \{v\}} \frac{w(u,v)}{\sqrt{\deg(v) \deg(u)}} W^{(\ell)} x_u^{(\ell-1)} \right) \tag{3}$$

where $W^{(\ell)}$ represents the learnable weights at layer $\ell$ and $w(u,v)$ is the weight of an edge $(u,v)$. Note that if there are no edge weights in the input graph, $w(u,v) = 1$. In contrast, graph attention networks (GATs) use sum-pooling to aggregate neighboring node features along with attention mechanisms within the aggregation function in order to assign levels of importance to node neighbors. At each layer $\ell$, GATs update node features as

$$x_v^{(\ell)} = \sigma \left( \sum_{u \in \mathcal{N}(v)} \alpha_{vu} W^{(\ell)} x_u^{(\ell-1)} \right), \tag{4}$$

where $\alpha_{vu}$ is a learnable attention coefficient. Edge features for this network are incorporated when computing the attention mechanism $\alpha_{vu}$ by concatenating the edge feature with the node feature before applying/computing the attention mechanism. Note that in our case, our input node features are the lift $x_v^{(0)} = \hat{v}$ for each $v \in V$ and input edge features, given an edge $(u,v)$, is the Euclidean distance between the lifts of $u$ and $v$.

## B. Transformers

### B.1. Transformer architecture

Transformers have become the state-of-the-art architecture for handling tasks in natural language processing and computer vision (Vaswani et al., 2017; Dosovitskiy et al., 2021; Liu et al., 2021). Each layer of the transformer is a *sequence-to-sequence permutation equivariant* function [2] and consists of a transformer block. Each transformer block includes a multi-head self-attention mechanism and a position-wise feed-forward neural network. Given a sequence $X \in \mathbb{R}^{n \times D}$, the output of a single attention head $\mathrm{Attn} : \mathbb{R}^{n \times D} \to \mathbb{R}^{n \times D}$ is computed as

$$\mathrm{Attn}(X) = \mathrm{softmax}(XW_Q(XW_K)^T)$$

where $W_Q, W_K \in \mathbb{R}^{D \times D}$. The output of this attention mechanism is then passed through an MLP which then computes the updated feature for each token independently so the final output is $\mathsf{MLP}(X + \mathrm{Attn}(X)XW_V)$ where $W_V \in \mathbb{R}^{D \times D}$. Intuitively, the $N \times N$ matrix $XW_Q(XW_K)^T$ computes self-similarity of any two elements in $X$ (although after linear transformations by $W_Q$ and $W_K$. Using the $\mathrm{softmax}$ normalizes the self-similarity values across each row, and therefore records, for each element in $X$, the "influence" ("attention") of any other element, and the multiplication with $XW_V$ then integrates such influences to update each element in $X$.

Since transformer parameters are shared across all elements of the input sequence, once the transformer is trained it can be applied to an input sequence of any size. While there have been several works adapting transformers for graph structured input (Dwivedi & Bresson, 2020), one can also directly apply standard transformers directly to graphs (Kim et al., 2022). In fact, one can view a transformer as a graph neural network over a complete graph where we compute an additional edge feature which captures the relative importance of pairs of nodes. However, the transformer takes quadratic time $O(|V|^2)$ whereas the standard message passing GNN is linear with respect to the size of the input graph $O(|V| + |E|)$.

### B.2. Intialization of embedding module with transformer

We also explore the use of a transformer as the initial embedding module $\phi$. In this approach, the entire graph is tokenized and presented as an input sequence, as described earlier in Section 2. The transformer treats the terrain graph as a global

---

[2]A sequence of $N$ elements, each being a $D$-dimensional vector, can be represented by a point $X \in \mathbb{R}^{ND}$. Intuitively, $X$ is a $N \times D$ matrix, where each of the $N$ rows is a $D$-dimensional vector. A function $f : \mathbb{R}^{ND} \to \mathbb{R}^{ND'}$ is *permutation equivariant* if its output respects the permutation of the rows of the input. In other words, for any $N \times N$ permutation matrix $\Pi$, we have that $f(\Pi X) = \Pi f(X)$.

| | Latent embedding dimension | | | | |
|---|---|---|---|---|---|
| $a$ | 4 | 16 | 32 | 64 | 128 |
| 2.0 | $3.63 \pm 2.84$ | $1.32 \pm 1.11$ | $1.04 \pm 1.16$ | $0.82 \pm 1.19$ | $1.27 \pm 1.70$ |
| 4.0 | $6.46 \pm 4.98$ | $2.93 \pm 3.34$ | $1.69 \pm 2.50$ | $1.95 \pm 3.09$ | $1.96 \pm 3.33$ |
| 10.0 | $9.41 \pm 8.57$ | $3.72 \pm 4.59$ | $2.77 \pm 3.81$ | $2.86 \pm 6.11$ | $2.87 \pm 7.16$ |

*Table 6.* Relative error ($\%$, $\downarrow$) versus latent embedding dimension for GAT + $L_1$ for the synthetic dataset with Gaussian amplitude $a \in \{2.0, 4.0, 10.0\}$. We see that relative error is mostly stable after embedding in $\mathbb{R}^{64}$.

information, where information from all nodes are simultaneously accessible during the embedding computation. Unlike when the GNN is used as the $\phi$, which explicitly leverages local connectivity to compute embeddings, the transformer operates on a global scale and can capture long-range dependencies. In short, transformers excel at incorporating global context at the cost of not necessarily being able to explicitly leverage local graph structure, whereas GNNs are more naturally suited for encoding local connectivity patterns but may struggle to capture long-range dependencies effectively, as mentioned previously.

While our discussion focuses on the single-terrain case, it is important to note that both the GNN and transformer-based approaches can naturally extend to cross-terrain settings. These models take the terrain graph as input, allowing them to generalize to multiple terrains without requiring a separate model for each. In contrast, the MLP-based approach is limited in this regard. Since the MLP processes input features independently and lacks any inherent mechanism to incorporate graph structure, a single MLP model is specific to a single function. Applying an MLP to cross-terrain settings would require retraining or maintaining a separate model for each terrain.

## C. Additional experimental details

### C.1. Training details

We train all models using PyTorch and 8 NVidia A1000 GPUs. As a note on hyperparameter tuning: in general, we tuned the depth (between $\{2, 3, 4\}$ layers) and output latent embedding of the initial embedding module $\phi$ (between $\{4, 16, 32, 64, 128\}$) on the small synthetic datasets. The best hyperparameters are then used to training the networks on the real-world terrain datasets. We focus on tuning the hyperparameters for Siamese embedding module and fix the final distance computation MLP hyperparameters at two layers and 256 hidden units. We make a special note of the dimension of the latent embedding for the Siamese GNN module as this is likely to affect the final quality of the Siamese network approaches and report the relative error for the Siamese network approach (GAT + $L_1$) on various synthetic terrains as the latent embedding dimension increases in Table 6. We note that more detailed results showing the relative error of all approaches with different latent embedding dimensions can be seen in Appendix C.3, Figure 5. Observe that the error incurred by the Siamese network stabilizes after embedding to $\mathbb{R}^{64}$. Therefore, for each GNN-based model, we utilize an embedding module with three GNN layers each with 128 hidden units and an output embedding dimension of 64. An exception is when training the full GAT+$L_1$ model on the Norway dataset where resource constraints necessitate training a smaller embedding module model configured with an output embedding dimension of 16 and 32 hidden units per layer. We use the same configuration for MLP and Transformer-based models and use two attention heads for the Transformer-based model. Note that in the case of M-CTR training on Norway, we first train on a downsampled terrain, which allows us to maintain the original configuration with a 64-dimensional output embedding. Each neural network is trained using the AdamW (Loshchilov & Hutter, 2019) optimizer with a learning rate of 0.001. Finally, each model is trained for 500 epochs except for GAT+$L_1$ trained on the full Norway, Holland, and Philadelphia datasets, which are trained on 100 epochs because of time considerations. In the final finetuning of the distance computation module (in our de-coupled/M-CTR training approach), we train the final MLP for 1000 epochs.

#### C.1.1. DATASET DETAILS

A complete table of train and test set sizes is provided in Table 7. As described in Section 3, $S_{train,1}$ serves as the initial training set in our decoupled approach, used to train the embedding module $\phi$, while $S_{train,2}$ is used to fine-tune the final distance computation MLP.

Since the artificial terrains are significantly smaller, we construct both $S_{train,1}$ and $S_{train,2}$ by randomly sampling 50K

source and target nodes along with their shortest path lengths. However, generating 50K random source-target pairs is computationally expensive for real terrains, even in their downsampled versions. To address this, we sample 1000 random source points per real terrain. For $S_{train,1}$, we select 500 targets per source, and for $S_{train,2}$, we sample 3500 targets. For the experiments with terrain uncertainty models, we reduce $|S_{train,2}|$ to accelerate training to ensuring that finetuning all 50 MLPs was computationally feasible. Specifically, we sample 500 targets per source for these models.

For the artificial terrains, we generate the test sets by sampling 1000 sources and taking their shortest path distances to every other node in the terrain. For the real terrains, we generate test sets by sampling 1000 source points and then sampling 500,000 targets across the terrains.

| | $|S_{train,1}|$ | $|S_{train,2}|$ | Test |
|---|---|---|---|
| Artificial | $5 \times 10^4$ | $5 \times 10^4$ | $2.5 \times 10^5$ |
| Norway-250 | $5 \times 10^4$ | $3.5 \times 10^5$ | $5 \times 10^8$ |
| Norway | $5 \times 10^4$ | $3.5 \times 10^5$ | $5 \times 10^8$ |
| LA | $5 \times 10^4$ | $3.5 \times 10^5$ | $5 \times 10^8$ |
| Holland | $5 \times 10^4$ | $3.5 \times 10^5$ | $5 \times 10^8$ |
| Philadelphia | $5 \times 10^4$ | $3.5 \times 10^5$ | $5 \times 10^8$ |
| Uncertainty models | $5 \times 10^4$ | $5 \times 10^4$ | $5 \times 10^8$ |

*Table 7.* Dataset sizes for all terrains. Note that Norway-250 refers to the downsampled 250x250 version of Norway used in Section 4 Table 2 for comparison with GeGNN.

### C.2. Time to generate initial embeddings

In addition to the time required for SP-distance approximation at inference, we also measure the time needed to compute each embedding for the GNN-based module in Table 8 (as GNNs were shown to have the best performance among all tested models). Since a single GNN pass scales linearly with the number of nodes and edges, $O(|V| + |E|)$, the embedding computation time increases with terrain graph size, as expected. However, once computed, embeddings can be stored and reused for distance queries, offering significant time savings—especially compared to the non-neural SOTA approach, which required several days of pre-processing time to construct a data structure for a terrain with 180,000 nodes (Wei et al., 2024).

| # of nodes | 625 | 2500 | 62500 | 4M | 16M |
|---|---|---|---|---|---|
| Time (s) | 0.001 | 0.5 | 2 | 21 | 107 |

*Table 8.* Time required to generate embeddings using a GAT with the specified hyperparameters above for input graphs of varying sizes. As the number of nodes increases, the embedding computation time scales accordingly.

### C.3. More results on exploring the design space on artificial terrains

In Figure 5, we show the average relative error of different model designs over the test sets for artificial terrains. We plot the average relative error ($y$-axis) against the 'complexity' (the maximum amplitude of the Gaussians) of each artificial terrain in the $x$-axis. We observe that GNN-based embedding modules consistently outperform MLP and transformer-based modules by a significant margin. This result is expected as MLP-based embedding methods, as discussed in Section 3, rely solely on pointwise information for training and cannot incorporate the terrain structure. Additionally, while transformers can encode global terrain information via the initial position encoding (of coordinates) as well as global self-attention operations, the self-attention mechanism does not appear to be as effective as GNN message-passing in retrieving path information. Indeed, it have been previously observed that GNNs naturally align with classical shortest-path algorithms like Bellman-Ford and Dijkstra, enabling them to better capture and propagate graph structure in a way that facilitates accurate SP-distance estimation (Xu et al., 2019; Veličković et al., 2020). We also note that transformers require $\Theta(|V|^2)$ time to compute self-attention, while GCNs and GATs are linear $O(|V| + |E|)$ to the size of input graph.

Furthermore, among the GNN choices, we observe that the GAT consistently performs better than the GCN. This again might not be surprising as GAT allows attending to different neighbors differently when aggregating messages at a graph node. We also note that GeGNN of (Pang et al., 2023) is worse than GCN+MLP (which can be viewed as the original

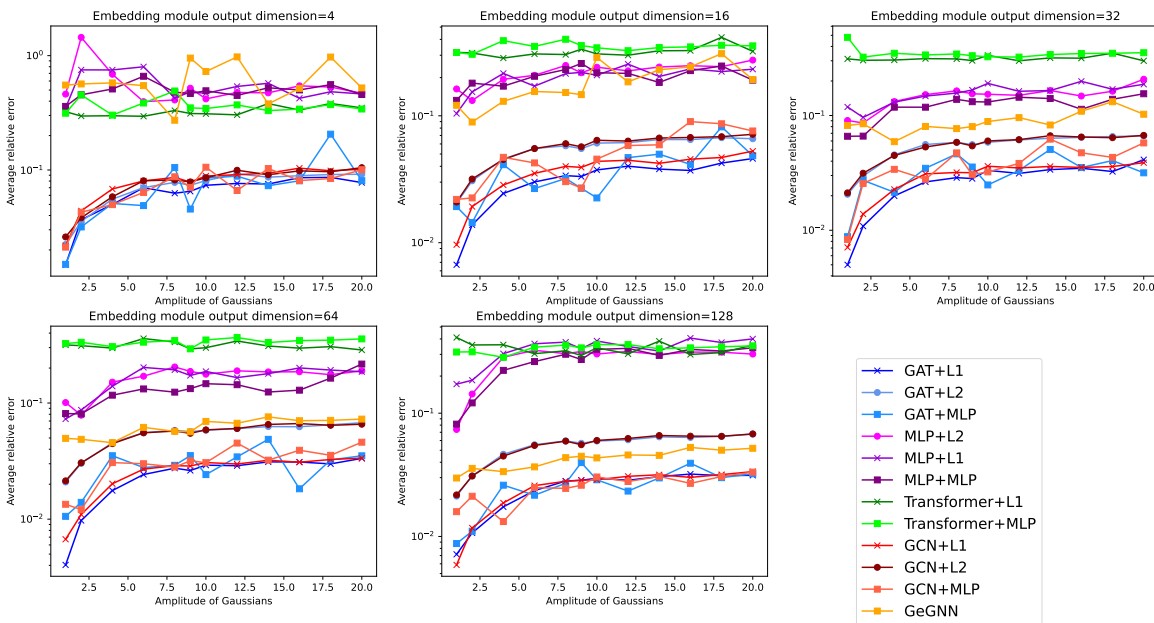

*Figure 5.* Average relative error for each model on artificial terrains. Each plot corresponds to a different output embedding size for $\phi$. The x-axis of each plot corresponds to the amplitude of the Gaussians in a given artificial terrain and the y-axis corresponds to the average relative error.

NeuroGF (Zhang et al., 2023)). Most interestingly, we note that using only $X+L_1$ consistently outperforms $X+MLP$ as well as $X+L_2$. From these observations, we narrow down our architectural explorations and exclude MLP and Transformer based architectures from our further experiments.

### C.4. Synthetic perturbations to terrain edge weights

We artificially perturb the terrain graphs of downsampled versions of Norway and Los Angeles (each of them 250x250 terrains) and introduce perturbations by scaling each edge weight by some value sampled from $\mathcal{N}(1.0, 1.0)$. Instead of retraining the $GAT + L_1$ embedding module, which is expensive, we can simply use our de-coupled training approach and re-train the final distance computation module according to the new weighted terrain. The relative error of the previously trained $GAT + L_1$ and the de-coupled approach is shown in Table 9. This updated de-coupled model achieves significantly lower error compared to directly using the previously trained model. While the original model cannot approximate shortest paths at all on the perturbed terrain, retraining just the distance computation module yields accurate shortest path estimates with minimal computational overhead.

|  | Norway | Los Angeles |
| --- | --- | --- |
| $GAT + L_1$ | 132% | 186% |
| updated-de-coupled | 3.46% | 4.36% |

*Table 9.* Average relative error on perturbed versions of $250 \times 250$ downsampled version of each node. We compare the decoupled training approach, where $\phi$ is first trained on a downsampled version of the terrain and then a final distance adjustment MLP is re-trained on the new terrain, against original $\phi$, which directly uses the $L_1$ distance between embeddings as the final estimate.

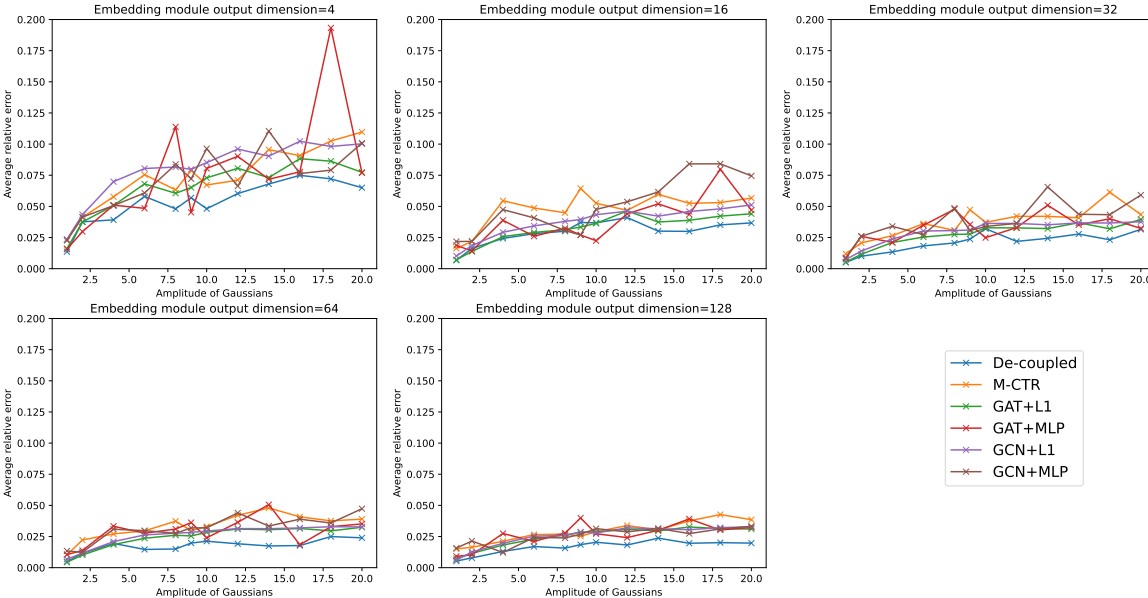

*Figure 6.* Expanded version of Figure 5 for GNN-based approaches only. We plot the average relative error (y-axis) of each GNN based model against the the amplitude of the Gaussians ('complexity') in an artificial terrain. Each plot corresponds to a different latent space dimension for the embedding module $\phi$.

