# OpenReview forum: "De-coupled NeuroGF for Shortest Path Distance Approximations on Large Terrain Graphs"
_ICML.cc/2025/Conference — ICML 2025 poster_

### Official Review · Reviewer_j9PW · 2025-03-10

**Overall Recommendation:** 5

**Summary:**

This paper proposes a new learning-based approach for answering shortest path distances on large-scale terrain DEMs. Overall, the proposed method extends the prior work of NeuroGF while providing a comprehensive and in-depth analyses on the training mechanisms and design choices of neural components. Extensive experiments demonstrate that the proposed method brings obvious improvement in terms of both accuracy, efficiency, and scalability.

**Claims And Evidence:**

Well supported.

**Essential References Not Discussed:**

Reference adequate.

**Experimental Designs Or Analyses:**

Well-organized experimental setup.

**Methods And Evaluation Criteria:**

Comprehensive experimental evaluations.

**Other Comments Or Suggestions:**

N/A

**Other Strengths And Weaknesses:**

In general, this paper deals with an essential and highly valuable problem of developing efficient geodesics answering frameworks, which are not fully investigated in the current community, especially for "neuralized" design paradigms. The ways of analyzing the working mechanisms and restrictions of existing baselines and further exploring more effective structural designs are solid and inspiring.

Particularly, the decoupled training of embedding and distance adjustment modules are technically sound. Since the first stage is performed on coarsen graphs, the overall training efficiency can be greatly improved when facing large-scale data. More importantly, we can circumvent the cumbersome re-training process by only fine-tuning the second stage.

In summary, I tend to think the proposed method is an insightful work to the problem of neural geodesics learning.

**Questions For Authors:**

In the right column of page 1, the authors mentioned "Throughout this paper we assume that a terrain is represented as a xy-monotone triangulated surface Σ in R^3". What does it mean specifically?

**Relation To Broader Scientific Literature:**

Closely related to geometry processing and downstream applications of geospatial data processing.

**Theoretical Claims:**

Technically sound.

---

> ### Author Rebuttal · Authors · 2025-03-31
>
> Thank you for your review! We are glad that you appreciate our proposed advancement in the problem of neural data structures for SP queries on terrains.
>
> **Regarding your question about $xy$-monotone surfaces**, a continuous surface in $\mathbb{R}^3$ is called $xy$-monotone if every line parallel to the $z$-axis only intersects the surface at a single point.

---

> > ### Comment · Reviewer_j9PW · 2025-04-08
> >
> > Thanks for the authors' explanations. I have no further concerns.

---

> > > ### Author Response · Authors · 2025-04-09
> > >
> > > Thank you for your thoughtful evaluation of our paper!

---

### Official Review · Reviewer_2aDC · 2025-03-13

**Overall Recommendation:** 4

**Summary:**

This paper proposes a De-coupled NeuroGF for efficiently approximating SPD queries on large-scale DEMs. The key contribution authors decouples the Siamese embedding module and the distance calculation module in NeuroGF. By combining an efficient two-stage hybrid training strategy, the method significantly reduces computational bottlenecks, making training on large-scale terrain DEMs (up to 16 million nodes) feasible. Experimental results demonstrate that the method performs excellently on both synthetic and real datasets.

**Claims And Evidence:**

The main claim is the decoupled training framework, combined with a two-stage hybrid training strategy, provides an efficient solution for SPD queries on large-scale terrain DEMs. This claim is supported by clear by experiments.

**Essential References Not Discussed:**

The comparison methods in the paper are only up to 2023, which may be somewhat outdated for ICML 2025. Are there any updated methods from 2024?

**Ethics Expertise Needed:**

["Other expertise"]

**Experimental Designs Or Analyses:**

I have read the experimental section and believe that it provides sufficient evidence to support the paper's claims.

**Methods And Evaluation Criteria:**

The evaluation criteria include mean relative error and accuracy, which are commonly used metrics for SPD approximation problems and are both reasonable and standard. The datasets including synthetic and real-world terrains, fully test method performance.

**Other Comments Or Suggestions:**

1.	It is recommended to include a sensitivity discussion or ablation study on key hyperparameters (such as embedding dimension, number of GNN layers, and coarsening factor) to improve reproducibility.

2.	compare with more recent methods.

**Other Strengths And Weaknesses:**

Strength:

1.	The paper is generally well-written, well-structured, and easy to read and understand.

2.	The proposed method achieves training on large-scale terrain DEMs and accelerates the training stage. This is the main advantage of the proposed method and is a good improvement.

3.	Extensive experiments on both synthetic and real datasets provide strong evidence.

Weakness:

1.	The paper does not provide detailed information on hyperparameter tuning or sensitivity analysis, which could affect the understanding of reproducibility and robustness. For example, in M-CTR, increasing the value of k reduces training time, but what impact does this have on accuracy? Does each dataset require separately designed parameters?

2.	The comparison methods are only up to 2023. Are there any methods from 2024 or more recent ones?

**Questions For Authors:**

Dividing the network into two parts to accelerate training is a common approach in the computer vision (CV) field, such as in ACE[1]. What are the advantages and differences of the method proposed by the authors?

[1] Brachmann E, Cavallari T, Prisacariu V A. Accelerated coordinate encoding: Learning to relocalize in minutes using rgb and poses[C]//Proceedings of the IEEE/CVF Conference on Computer Vision and Pattern Recognition. 2023: 5044-5053.

**Relation To Broader Scientific Literature:**

This paper refers to NeuroGF, and extend this idea to large-scale terrain DEMs, addressing the scalability challenges

**Theoretical Claims:**

I think this paper focus on the practical implementation. Therefore, there are no theoretical claims or proofs that need to be verified.

---

> ### Author Rebuttal · Authors · 2025-03-31
>
> We thank you for your time and constructive feedback. We are glad the reviewer appreciates our de-coupled and mixed coarse-to-refined training strategy for efficiently processing large terrain graphs at scales previously which were not able to be considered. As you point out, our de-coupled training strategy is particularly useful for dynamically updating neural data structures efficiently.
>
> **Hyperparameter tuning:** Thank you for these comments. We should have included such information in the paper. In fact, we do deploy hyperparameter tuning in the paper. For our experiments, we perform all hyperparameter tuning on smaller-scale synthetic terrains, selecting the best-performing hyperparameters based on these trials. These optimized hyperparameters are then applied across all subsequent experiments on large-scale terrain graphs. We will include detailed hyperparameter tuning in the revision. An example of how the model accuracy changes with different embedding dimensions for the latent space is shown in Figure 1 (https://shorturl.at/SXyB2). In general, we observe that model performance stabilizes after a while.
>
> **Comparisons with other work:** The primary goal of our paper is to develop a lightweight neural data structure to answer shortest-path (SP) distance queries for extremely large terrain graphs. We see three relevant areas of work for comparison:
>
> (1) Metric learning: Siamese networks are the SoTA architecture in the field of metric learning and we already compare with state-of-the-art Siamese network, denoted by $\mathsf{X}$+$L_p$ in our paper. Furthermore, we explored the design space of Siamese learning approaches by using different SoTA GNN architectures and transformers used as the network backbone. (See C.3 in the Supplement.)
>
> (2) Neural models specific for geodesic distance queries: GeGNN and NeuroGF are the current SoTA architectures for processing geodesic distance queries (i.e. shortest path distance) on graphs induced by meshes. In our paper, we provide direct comparisons to both GeGNN and NeuroGF.
>
> (3) General graph learning: While SPD approximation is a commonly considered problem in the graph learning community, our goals are different as we seek a lightweight and efficient neural data structure. The setting here is that once we preprocess the data (i.e. trained the model), the users might have many future SP queries, each of which consists of two points, and the model should return the SP distance between them quickly. (This is a fundamental primitive in GIS applications.)
> Current GNNs are not directly suitable as a neural data structure for answering many future SP queries. For example, there are several existing recurrent GNN methods which imitates algorithmic control flows in order to generalize all graph instances (Tang, et al. 2020, Luca, et al. 2024). However, such iterative approaches are computationally expensive during the inference time for each SP distance query, as it requires many iterations of the models to essentially explore the entire graph. In contrast, our latent embedding can capture hidden patterns in the SP distance function, while MLP can effectively retrieve the final distance. As an example, we trained a lightweight GNN model trained to learn two steps of single-source shortest paths. The inference time for approximating the SPD between a pair of nodes with 100 iterations of our lightweight model is 307 seconds on Norway as such a model is essentially forced to compute all shortest paths on the graph from a single source even though we are **only** interested the SP distance between a single pair. In contrast, our M-CTR method, after computing all initial embeddings, achieves an inference time of merely $7 \times 10^{-6}$ seconds. We would be happy to add such a comparison to the revised paper.
>
> In short, we have compared our method with all relevant neural approaches; but we would be happy to add additional comparisons with iterative GNNs. Note that all neural approaches are orders of magnitude more efficient than SoTA classical algorithmic approaches (as we described in paper).

---

> > ### Comment · Reviewer_2aDC · 2025-04-08
> >
> > Thank you to the author for the detailed response, which has addressed my concerns. I agree with reviewer j9PW's assessment that this is a meaningful piece of work, and I have decided to increase my score.

---

> > > ### Author Response · Authors · 2025-04-09
> > >
> > > Thank you for your thoughtful evaluation of our paper!

---

### Official Review · Reviewer_Qpau · 2025-03-15

**Overall Recommendation:** 3

**Summary:**

This paper presents decoupled-NeuroGF framework for efficient approximate SPD queries on large terrain DEMs, based on the NeuroGF framework. This paper appropriately abstracts high-resolution terrain datasets as weighted graphs. The proposed decoupled-NeuroGF with a two-stage mixed-training strategy significantly improves computational efficiency and model performance, making the method suitable for large scale terrain. It reduces training time from days to hours for million-scale terrains and maintains high SPD approximation accuracy. The method also supports efficient updates for terrain changes through retraining only the distance-adjustment module. Experiments on real-world terrains with up to 16 million nodes demonstrate its scalability and superiority over previous approaches.

**Claims And Evidence:**

The paper's claims are well - supported by evidence:

[1] The significant reduction in training time is evidenced by experiments showing the method cuts training from days to hours for large terrains.

[2] Model performance improvement is proven by data indicating the method maintains high SPD approximation accuracy and enables efficient updates for terrain changes.

[3] The two - stage mixed - training strategy's benefits are experimentally validated, proving its effectiveness in enhancing computational efficiency and model performance.

To strengthen the evidence:

More comparative data on terrains of different scales could be provided to better demonstrate the method's versatility and efficiency across various terrains.

**Essential References Not Discussed:**

No essential references appear to be missing from the paper's discussion.

**Experimental Designs Or Analyses:**

The paper's experimental design for SPD queries on large-scale terrain DEMs is reasonable. It uses synthetic terrains with varying complexity but the same size, generated via 2D Gaussian mixtures, to evaluate different model designs. However, evolving more datasets with more vertices and corresponding complexity is recommended, as this paper aims to solve SPD question on high-resolution terrain datasets.

**Methods And Evaluation Criteria:**

The methods and evaluation criteria presented in the paper are reasonable for the problem of efficient SPD queries on large-scale terrain DEMs. The decoupled-NeuroGF data structure and the two-stage mixed-training strategy address computational bottlenecks and enable efficient training on large terrains. The evaluation focuses on key aspects like training time reduction, approximation accuracy, and efficient updates for terrain changes, which are appropriate for assessing the method's effectiveness in real-world applications.

**Other Comments Or Suggestions:**

No additional comments.

**Other Strengths And Weaknesses:**

Strengths.:
1. The proposed decoupled-NeuroGF framework represents a creative advancement in the field. By separating the training process into two stages, it effectively reduces computational demands and enhances training efficiency on large-scale terrains, which is a significant improvement for practical applications.
2. The method's ability to handle dynamic terrains by updating only the distance-adjustment module is also a valuable contribution, as it addresses the realistic challenge of terrain changes.

Weaknesses:
1. The experimental section could be strengthened by including more diverse and large-scale datasets to better showcase the method's generalizability.
2. While the paper presents a novel approach, a more detailed comparison with other existing methods would help better position its contributions within the broader literature.
3. The scope of the work appears to be somewhat limited. The paper could benefit from a more extensive exploration of the framework's capabilities and potential applications, which would provide a more comprehensive understanding of its value and impact.

**Questions For Authors:**

High-resolution terrain data set is a fundamental assumption in this paper.

[1] How to determine whether a data set belongs to this category?

[2] Could some indicators like vertex density be used for quantification? The paper seems to lack relevant analysis.

**Relation To Broader Scientific Literature:**

The key contributions of this paper are closely related to the broader scientific literature in several ways:

[1] Extension of NeuroGF Framework: This paper builds upon the existing NeuroGF framework, which is already a significant contribution to the field of geodesic distance estimation. By proposing a decoupled-NeuroGF data structure, the authors extend the applicability and efficiency of the original framework, addressing its limitations in handling large-scale terrain DEMs.

[2] Innovative Training Strategy: The two-stage mixed-training strategy is an advancement in training neural data structures for terrain analysis. This approach not only reduces computational bottlenecks but also allows for efficient training on large terrains, which was previously challenging. This strategy can be seen as a progression from traditional training methods, incorporating insights from model optimization and efficient learning techniques.

**Theoretical Claims:**

The theoretical claims regarding complexity in the paper are correctly proven.

---

> ### Author Rebuttal · Authors · 2025-03-31
>
> We thank the reviewer for your time and constructive feedback. We are happy that the reviewer appreciated our innovative training strategy: with our de-coupled and mixed coarse-to-refined training strategy, we introduce a lightweight neural data structures that can efficiently answer many shortest path distance (SPD) queries over massive terrain graphs orders of magnitude faster than classic algorithmic approaches. Note that this is a fundamental problem in GIS (see also our discussion in “Scope of work”). Below we provide responses for main questions/comments.
>
> **High-resolution terrain datasets:**  In our experiments, we use high-resolution terrain graphs because the shortest paths on these graphs provide more accurate approximations of the true geodesic on the terrain surface (and as a result, these are the most common data available). However, our technique is applicable to any terrain graph, including lower resolution ones where the geodesic approximation is coarser.
>
> We think that the resolution of the terrain graph does not directly affect the quality of the approximation; rather, the ``complexity" of the metric induced over the terrain graph would. As an example, we measure complexity by the _doubling dimension_ of the terrain graph. This is theoretically motivated by Theorem 1.2 of (Naor et al. 2012), which states that every metric space can be approximately embedded into $R^N$ with distortion dependent on the _doubling dimension_ of the original metric space. We conduct new experiments on synthetic terrains (see Table 1). We observe that: (1) as the doubling dimension increases, the relative error incurred by the Siamese embedding approach (i.e, GAT+$L_1$) also increases. (2) Our de-coupled training approach can help to adjust the errors from the Siamese approach (via the distance-computation module) and further improve the SPD approximation. Also the relative error by our de-coupled approach increases at a slower rate as doubling dimension increases, compared to that of the Siamese approach (GAT+L1).
> Theoretically, it could be interesting to provide sample complexity bounds w.r.t. doubling dimension but we leave this to future work.
> | Doubling Dimension| De-coupled| GAT + L |
> |-|-|-|
> | 4.1| **0.0052**| 0.0061|
> | 4.4| **0.0132**| 0.0166|
> | 4.7| **0.0186**| 0.0329|
> *Table:  Approximate doubling dimension, and average relative error of each model.*
>
> **Comparisons to other methods:** Please see our response to Reviewer 2aDC.
>
> **Experiments with more diverse and large-scale datasets:** Thank you for the suggestion. First, we note that we focus on Norway and Los Angeles in our experiments as are they are both complex and large-scale. Both Norway (4M nodes) and Los Angeles (16M nodes) are far larger than any terrain graphs considered in previous work. They both represent highly complex terrains: The Norway terrain graph is taken from a highly mountainous region and the Los Angeles dataset spans all of LA county and has both flat and mountainous regions with elevations varying between 3000m and 0m.
>
> We will include more large-scale datasets in the revision. We have already obtained results for two additional datasets: Holland and Philadelphia, both of which contain 1M nodes. Results are shown in Table 2. For both datasets, our new M-CTR approach outperforms just simply using the Siamese network trained on a coarse 2500 node version of the terrain (Coarse GAT+L1). We will update our paper with these results, as well as additional results from larger datasets (e.g. a 25M node Norway dataset which covers a different region of Norway than our current Norway dataset).
> | Dataset| Model| Relative Error (%) ↓ | Accuracy (%) ↑ |
> |-|-|-|-|
> |**Holland, IN**| Coarse GAT + L1| 2.06 ± 1.54| 65.1|
> | | M-CTR| **0.86 ± 2.29** | **90.8**|
> | **Philadelphia, PA**| Coarse GAT + L1| 2.07 ± 1.47| 30.1|
> |  | M-CTR| **0.51 ± 0.71** | **94.4** |
>
> **Scope of the work:** First, we would like to emphasize that the development of succinct data structures to support efficient SP distance queries on massive terrain graphs is important in its own right: it is a fundamental problem in geospatial data analysis and GIS database systems with a wide range of applications, as SPD queries is a key primitive operation from terrain-navigation, to point of interest search problems, to flood simulations. Thus a scalable, practical data structure for SP-distance queries on terrains will have a huge impact. This explains the huge literature on this problem both in GIS and computational geometry. We also show that our two-stage mixed training strategy leads to an easy-to-update neural data structure when there are dynamic changes in the terrain (e.g. in time-sensitive natural disaster relief scenarios). Furthermore, while not explored in this paper, our de-coupled training framework could also be applicable to other general metric learning setups, when the input metric space is massive, and a large number of future SP queries are expected.

---

### Decision · Program_Chairs · 2025-05-01

**Decision:**

Accept (poster)

**Comment:**

This paper presents a novel decoupled neural geodesic field framework and an efficient training strategy for approximating shortest-path distances on large-scale terrain digital elevation models. All three reviewers with scores ranging from 3 to 5, recommend accepting the paper. I believe this paper can be a good addition to the ICML 2025. The authors are encouraged to incorporate the reviewers' constructive feedback into the final camera-ready version.